# Prostate Cancer Treatments and Their Effects on Male Fertility: Mechanisms and Mitigation Strategies

**DOI:** 10.3390/jpm15080360

**Published:** 2025-08-07

**Authors:** Aris Kaltsas, Nikolaos Razos, Zisis Kratiras, Dimitrios Deligiannis, Marios Stavropoulos, Konstantinos Adamos, Athanasios Zachariou, Fotios Dimitriadis, Nikolaos Sofikitis, Michael Chrisofos

**Affiliations:** 1Third Department of Urology, Attikon University Hospital, School of Medicine, National and Kapodistrian University of Athens, 12462 Athens, Greece; ares-kaltsas@hotmail.com (A.K.); nikosrazos@gmail.com (N.R.); zkratiras@med.uoa.gr (Z.K.); ddelijohn@med.uoa.gr (D.D.); stamarios@yahoo.gr (M.S.); constantinos.adamos@gmail.com (K.A.); 2Department of Urology, Faculty of Medicine, School of Health Sciences, University of Ioannina, 45110 Ioannina, Greece; azachariou@uoi.gr (A.Z.); nsofikit@uoi.gr (N.S.); 3Department of Urology, Faculty of Medicine, School of Health Sciences, Aristotle University of Thessaloniki, 54124 Thessaloniki, Greece; difotios@auth.gr

**Keywords:** prostate cancer, male infertility, sperm cryopreservation, assisted reproduction, testicular damage, androgen deprivation therapy, ejaculatory dysfunction, DNA fragmentation, fertility preservation, survivorship

## Abstract

Prostate cancer (PCa) is the second most frequently diagnosed malignancy in men worldwide. Although traditionally considered a disease of older men, the incidence of early-onset PCa (diagnosis < 55 years) is steadily rising. Advances in screening and therapy have significantly improved survival, creating a growing cohort of younger survivors for whom post-treatment quality of life—notably reproductive function—is paramount. Curative treatments such as radical prostatectomy, pelvic radiotherapy, androgen-deprivation therapy (ADT), and chemotherapy often cause irreversible infertility via multiple mechanisms, including surgical disruption of the ejaculatory tract, endocrine suppression of spermatogenesis, direct gonadotoxic injury to the testes, and oxidative sperm DNA damage. Despite these risks, fertility preservation is frequently overlooked in pre-treatment counseling, leaving many patients unaware of their options. This narrative review synthesizes current evidence on how PCa therapies impact male fertility, elucidates the molecular and physiological mechanisms of iatrogenic infertility, and evaluates both established and emerging strategies for fertility preservation and restoration. Key interventions covered include sperm cryopreservation, microsurgical testicular sperm extraction (TESE), and assisted reproductive technologies (ART). Psychosocial factors influencing decision-making, novel biomarkers predictive of post-treatment spermatogenic recovery, and long-term offspring outcomes are also examined. The review underscores the urgent need for timely, multidisciplinary fertility consultation as a routine component of PCa care. As PCa increasingly affects men in their reproductive years, proactively integrating preservation into standard oncologic practice should become a standard survivorship priority.

## 1. Introduction

Prostate cancer (PCa) is among the most prevalent malignancies in men, with approximately 1.4 million new cases diagnosed globally in 2020, ranking as the second most frequently occurring cancer in the male population worldwide. Although the incidence remains highest in older adults—peaking in the seventh decade of life—recent epidemiological trends reveal a gradual shift toward younger age at diagnosis. Specifically, early-onset PCa (defined as diagnosis before age 55) is increasing in incidence, with an estimated annual rise of 2–3% globally [1,2]. In the United States, approximately 10% of new PCa cases now occur in men under 55 [3], and similar patterns are observed in Europe and other high-income regions [4]. This trend is attributed in part to expanded use of prostate-specific antigen (PSA) screening and advances in imaging and biopsy techniques—such as multiparametric MRI and MRI/ultrasound fusion-guided transperineal biopsy—which have improved detection of clinically significant disease in younger men [5,6]. Furthermore, there is growing recognition that male infertility can be a biomarker of general health and may signal elevated risks of chronic diseases, including prostate malignancy [7].

Concurrently, therapeutic advancements have dramatically improved disease-specific outcomes. Five-year survival rates for PCa now exceed 95% in most high-income countries, and long-term survival is common for localized and early-stage disease [8]. Consequently, the population of PCa survivors is expanding; in the United States alone, over 3.5 million men are living with a history of PCa [9]. As survivorship becomes a central focus of oncology care, there is increasing recognition that long-term quality-of-life domains—such as sexual function, psychological well-being, and achievement of reproductive goals—must be addressed proactively.

Among these concerns, male fertility has emerged as a significant yet under-recognized issue in PCa survivorship. A substantial proportion of men diagnosed in their 40s and early 50s have not completed childbearing or may desire future biologic fatherhood. However, curative treatments—including radical prostatectomy, pelvic radiotherapy, and long-term androgen deprivation therapy (ADT)—commonly result in irreversible infertility through mechanisms such as anejaculation, ejaculatory duct obstruction, and suppression of spermatogenesis [10]. Despite this reality, fertility preservation is often not addressed in PCa care pathways: many younger patients are not adequately counseled on the reproductive consequences of treatment and are not offered referrals for sperm banking or fertility consultation prior to therapy [11]. This gap in care highlights a critical unmet need.

Notably, few prior reviews or guidelines have comprehensively focused on fertility in the context of PCa, leaving a significant literature gap. While fertility preservation in younger cancer patients has been discussed broadly, specific considerations unique to PCa (such as surgical anejaculation and prolonged ADT effects) are underrepresented in the literature. The present review aims to fill this gap by providing a detailed, up-to-date synthesis of knowledge on PCa treatment-related fertility impairment and by highlighting emerging opportunities to improve reproductive counseling and outcomes for these patients. In preparing this narrative review, we performed a comprehensive literature search (through early 2025) using databases such as PubMed and Scopus, with keywords including “prostate cancer,” “male infertility,” “fertility preservation,” “sperm banking,” and “oncofertility.” Relevant original research, review articles, and clinical guidelines were included, with emphasis on recent data and high-impact findings.

Given the shifting demographic and survivorship priorities in PCa, this review provides a timely overview of the impact of PCa treatments on male fertility, surveys available fertility preservation strategies, and discusses emerging technologies and unmet clinical needs in this evolving field. The goal is to empower clinicians with knowledge to integrate fertility considerations into PCa management and to inform future research directions in this important domain.

## 2. Prostate Function and Male Reproductive Health

The prostate gland is the largest male accessory reproductive gland and contributes roughly one-third of the total ejaculate volume [12]. Its secretions are biochemically rich and play indispensable roles in male fertility. Prostatic fluid contains exceptionally high concentrations of zinc and citrate, proteolytic enzymes such as prostate-specific antigen (PSA, a kallikrein-related serine protease), and nanosized membranous vesicles known as prostasomes [12,13]. These secretions not only provide structural and regulatory molecules for sperm function but also orchestrate key phases of the ejaculatory process—including semen liquefaction, sperm capacitation, and acrosome reaction priming—thereby underscoring the prostate’s pivotal role in coordinating male reproductive physiology [14].

One of the key enzymes produced by the prostate is PSA, a kallikrein-related serine protease (KLK3), which orchestrates post-ejaculatory semen liquefaction. Upon ejaculation, seminal vesicle proteins, mainly semenogelins and fibronectin, induce semen to coagulate into a gel-like matrix that temporarily immobilizes sperm [15]. Initially, PSA and related kallikreins are inhibited by zinc ions, which are abundant in seminal plasma. However, due to zinc’s preferential binding to semenogelins, zinc gradually dissociates from PSA, permitting its activation [16]. The active enzyme then cleaves the structural proteins of the coagulum, enabling sperm to regain motility and traverse the cervical mucus en route to the oocyte. This zinc-dependent mechanism of delayed semen liquefaction is critical for the temporal regulation of sperm release within the female reproductive tract [17].

Prostatic fluid contains remarkably elevated levels of zinc, which serves multiple, interrelated roles in coordinating the post-ejaculatory functionality of semen. It regulates the activation of PSA through competitive binding. It also stabilizes sperm chromatin by interacting with protamines, thereby preserving DNA integrity [18]. Zinc reinforces sperm membrane architecture and exerts antimicrobial properties against pathogens in the female reproductive tract [19]. In addition, it modulates the maternal immune response by inhibiting leukocyte activation. Elevated seminal zinc levels help maintain sperm in a non-capacitated state immediately after ejaculation, while gradual dilution along the female reproductive tract enables capacitation and subsequent fertilizing potential [20]. The high intraprostatic accumulation of zinc is made possible by androgen-sensitive zinc transporters, primarily ZIP1–4 and ZnT1–10, which coordinate zinc uptake and intracellular sequestration. This hormonally regulated zinc trafficking system contributes to both prostate function and fertility, and is markedly disrupted in prostate malignancies [21,22].

In parallel with zinc, citrate represents another major prostatic secretion with equally critical roles in seminal homeostasis. Citrate accumulation in the prostate is facilitated by a unique metabolic shift: prostatic epithelial cells suppress mitochondrial aconitase activity, effectively inhibiting the Krebs cycle [23]. This zinc-dependent metabolic block leads to the accumulation of high concentrations of citrate, which is then secreted into the prostatic fluid [23]. Citrate serves as a buffering agent that helps maintain the slightly alkaline pH of semen, which is vital for sperm viability in the acidic vaginal environment [24]. Furthermore, citrate provides an energy substrate that supports sperm motility during their journey toward the oocyte. This distinctive metabolic phenotype—marked by glycolytic ATP production and citrate accumulation—sets prostatic epithelial cells apart from all other healthy human tissues and underscores their highly specialized reproductive function [25]. The combined actions of zinc and citrate create a chemically favorable environment that enhances sperm longevity and fertilizing competence [26].

In addition to soluble molecules, the prostate secretes prostasomes—extracellular vesicles ranging from 50 to 200 nanometers in diameter [27]. These vesicles are enriched with proteins, lipids, and regulatory RNAs (e.g., microRNAs and long non-coding RNAs, which may modulate post-transcriptional gene expression relevant to sperm function) and contribute to distinct aspects of sperm physiology, including immunomodulation, membrane fusion, and motility enhancement [28]. Following ejaculation, prostasomes attach to the sperm surface and are transported through the female reproductive tract. Upon reaching more distal sites, prostasomes fuse with sperm membranes, delivering molecular cargo that enhances motility and facilitates the acrosome reaction—a crucial event allowing sperm to penetrate the zona pellucida of the oocyte [29]. Moreover, prostasomes exert immunomodulatory effects by dampening the activity of monocytes, neutrophils, and natural killer cells, thereby creating an immune-privileged environment for sperm survival [30].

The interplay of these secretions is essential not only for the immediate functionality of sperm but also for ensuring that fertilization can occur under the challenging physiological conditions of the female genital tract. Disruption of prostatic secretion—due to inflammation, obstruction, or surgical removal—has been associated with impaired semen quality, reduced sperm motility, and decreased fertilizing potential [31].

Understanding the intricate role of the prostate in male fertility provides a scientific foundation for integrating fertility preservation strategies into oncologic counseling, especially in the context of PCa survivorship and the development of personalized, fertility-preserving interventions [32].

## 3. Prostate Cancer Treatments and Their Impact on Fertility

PCa treatments encompass a range of surgical, radiotherapeutic, and systemic modalities, each capable of impairing male fertility through distinct and often multifactorial mechanisms. The reproductive consequences of these interventions vary in severity and reversibility, depending on the nature and extent of gonadal, hormonal, or ejaculatory disruption. Established modalities such as radical prostatectomy and pelvic radiotherapy exert well-documented adverse effects on fertility, primarily through anatomical excision or radiation-induced gonadotoxicity. In contrast, emerging systemic therapies introduce novel challenges, with limited clinical data necessitating cautious interpretation of their potential reproductive impact [33]. A comprehensive overview of fertility-related outcomes associated with each treatment modality is provided in Table 1.

### 3.1. Radical Prostatectomy

Radical prostatectomy (RP) involves complete surgical removal of the prostate gland and seminal vesicles, typically with transection of the vas deferens bilaterally. By design, this procedure directly eliminates antegrade ejaculation: virtually 100% of patients are rendered anejaculatory because the anatomical structures responsible for seminal fluid production and emission are excised or disrupted. While testicular sperm production may remain intact after RP, the loss of seminal vesicles and disconnection of the vasa deferentia result in aspermia during orgasm. Consequently, natural conception is impossible after RP unless sperm are retrieved and used with assisted reproduction [34,35].

Erectile dysfunction is also common post-RP (due to neurovascular bundle damage), though its degree depends on nerve-sparing status and patient age; erectile function per se is not necessary for sperm retrieval but is relevant for unassisted conception and psychosexual well-being [36]. On the hormonal side, RP has minimal direct impact—serum testosterone levels generally remain unchanged, though there may be a mild compensatory rise in gonadotropins (FSH/LH) due to testicular feedback [37].

Although ejaculation is generally considered universally lost after RP, a rare case of postejaculatory spermaturia has been reported following robotic-assisted prostatectomy, likely due to an iatrogenic fistulous connection between the vas deferens and the urinary tract [38]. While the sperm were immotile and the patient was infertile, the observation suggests that, in exceptional cases, inadvertent vasal continuity might persist or re-establish. Although such outcomes are unpredictable and not clinically actionable, they raise speculative interest in whether future surgical modifications might facilitate vasal continuity for assisted reproduction purposes.

### 3.2. Radiotherapy (External Beam and Brachytherapy)

Radiotherapy for PCa, while preserving the prostate anatomically, has significant adverse effects on male fertility [39]. Both external beam radiotherapy (EBRT) and brachytherapy expose the testes to some degree of scatter radiation, as well as directly impact the prostate and seminal vesicles. Clinically, ejaculatory dysfunction is common: long-term studies report that approximately 80–90% of men experience dry orgasm or markedly reduced semen volume after definitive radiotherapy [40]. This is due to radiation-induced damage to the prostate secretory tissue and ejaculatory ducts, as well as possible neuropathy affecting emission.

Sexual function may also be compromised following EBRT, with many patients reporting a decline in libido and sexual satisfaction. This phenomenon is likely multifactorial, involving subtle hormonal changes, post-treatment fatigue, and psychological distress. Although not as abrupt as with ADT, reduced testosterone levels and diminished sexual desire have been observed after EBRT in certain cohorts [41].

Additionally, radiation is highly gonadotoxic to the germinal epithelium in the testes. Even when spermatogenesis persists, ejaculate samples post-radiotherapy frequently demonstrate reduced sperm concentration, motility, and overall seminal volume, with a subset of patients developing oligozoospermia or azoospermia. Additionally, radiation can alter the biochemical composition of seminal plasma, rendering it less supportive of sperm function [42].

Furthermore, radiotherapy contributes to sperm DNA damage. Elevated levels of DNA fragmentation have been documented following both EBRT and brachytherapy, with clinical concern for reduced fertilization potential and increased risk of adverse reproductive outcomes [43]. As such, sperm DNA integrity should be considered a key aspect of post-treatment reproductive assessment. Given these cumulative effects, clinical guidelines strongly advocate for sperm cryopreservation prior to the initiation of curative radiotherapy [44].

### 3.3. Androgen Deprivation Therapy (ADT)

ADT is a cornerstone of treatment for advanced and high-risk PCa, whether as primary, neoadjuvant, adjuvant, or palliative treatment [45]. By inducing profound hypogonadism, ADT has a systemic impact on male fertility. The hallmark reproductive effects are a drastic reduction in libido, cessation of sexual activity, and severely impaired ejaculation [46]. Many patients report little or no fluid with orgasm after several months of ADT, indicating effectively zero sperm output in semen [47]. Importantly, ADT suppresses spermatogenesis: by lowering intratesticular testosterone via LH/FSH suppression, it leads to testicular atrophy and degeneration of the seminiferous epithelium [48].

In effect, most men become azoospermic or severely oligospermic while on ADT. Unlike surgery or radiation, ADT does not physically remove or irradiate germ cells, so recovery of sperm production might occur after cessation of therapy—but this is highly variable. Clinical experience shows that younger men on short-term ADT (≤6–12 months) may recover testosterone levels and spermatogenesis within 6–12 months after stopping ADT [49]. In contrast, older patients or those on prolonged (multi-year) ADT often have incomplete or no recovery of fertility, with persistent azoospermia or subfertility even after testosterone normalization [50].

Given this unpredictability, patients must be counseled about the high likelihood of infertility—even if given temporarily—and the possibility that fertility may not return after treatment, especially for extended ADT courses [51]. Moreover, in cases where ADT is combined with pelvic radiotherapy, cumulative gonadotoxicity may further diminish the likelihood of fertility recovery [52].

### 3.4. Chemotherapy

Cytotoxic chemotherapy (e.g., docetaxel, cabazitaxel, mitoxantrone, or combination regimens like docetaxel–estramustine) is used in certain high-risk or metastatic PCa settings [53]. Such agents target rapidly dividing cells and do not spare the spermatogenic epithelium, making them markedly gonadotoxic. During chemotherapy, men typically develop severe oligospermia or azoospermia, and even after completing treatment, sperm counts can take a long time to recover—if they recover at all [54]. For example, studies of taxane-based chemotherapy observed sharp declines in serum inhibin B and rises in FSH during treatment, accompanied by measurable reductions in testicular volume [55]. These hormonal changes reflect significant germ cell loss. In some cases, spermatogenesis may resume partially after many months or years once the chemotherapeutic insult is removed, but a substantial subset of patients remain persistently azoospermic or severely oligospermic in the long term [56]. A review of young cancer survivors found that 15–30% of men ultimately have permanent infertility after treatments like chemotherapy and radiotherapy [57]. Alkylating agents (not commonly used in front-line prostate cancer therapy, but sometimes in clinical trials or combination regimens) have the highest risk for permanent sterility [58].

Notably, chemotherapy generally does not cause immediate anatomical changes to the reproductive tract—the ejaculatory ducts and organs remain intact, and testosterone levels often return to normal after therapy (persistent hypogonadism is uncommon, though transient declines in T during therapy are observed) [59].

Therefore, some men maintain sexual function (erections and orgasm) post-chemotherapy, but the absence of viable sperm in the ejaculate precludes natural conception [60,61]. Because recovery of fertility after chemotherapy is highly unpredictable—some patients regain normal counts after a couple of years, whereas others remain infertile indefinitely—pre-treatment sperm cryopreservation is considered essential for any male patient receiving gonadotoxic chemotherapy [62]. This recommendation is supported by oncology guidelines and should be a standard part of pre-chemotherapy counseling in PCa patients of reproductive age.

### 3.5. Next-Generation Androgen Receptor Pathway Inhibitors

In recent years, novel hormonal agents (androgen receptor pathway inhibitors, ARPIs), such as enzalutamide, apalutamide, darolutamide, and the CYP17 inhibitor abiraterone, have become integral in managing advanced or high-risk PCa [63]. These next-generation AR therapies have unique implications for fertility. When used in combination with traditional ADT, the effects simply overlay on castrated testosterone levels—thus, EF is already absent and spermatogenesis suppressed due to ADT. However, some of these agents are also used as monotherapy in certain contexts (e.g., anti-androgen monotherapy) [64].

Limited human data exist on their direct impact on fertility, but animal studies and mechanistic insights suggest significant effects. Enzalutamide and similar AR antagonists, even given alone, can disrupt the hypothalamic–pituitary–gonadal axis: by blocking AR feedback, they may actually raise serum testosterone and gonadotropin levels (disinhibition of the pituitary), yet paradoxically reduce intratesticular testosterone and impair spermatogenesis [65]. In animal models, enzalutamide has been associated with reduced sperm counts and motility despite normal serum testosterone, attributable to this intratesticular testosterone decrease and direct effects on AR-dependent tissues [66]. Clinically, men on ARPIs commonly report decreased ejaculate volume and libido as side effects [67].

For abiraterone acetate, a potent inhibitor of androgen biosynthesis, profound suppression of both serum and intratesticular testosterone leads to a functional castration state, akin to ADT. This mechanism is expected to strongly impair spermatogenesis, though direct human fertility data are lacking [68]. Nonetheless, preclinical findings and regulatory guidance suggest that all ARPIs may adversely affect male reproductive function while on therapy [69,70].

Fertility recovery after stopping these drugs is presumed to be possible (since the drugs have finite half-lives and their effects on the axis should be reversible), but there have been no systematic studies on post-therapy paternity or time to sperm recovery. Given the uncertainties, it is prudent to treat ARPIs as potentially sterilizing while on treatment and counsel patients accordingly [71]. Sperm banking should be offered before initiating AR pathway inhibitors, especially in younger men or those on monotherapy, since we lack long-term follow-up data on fertility outcomes.

### 3.6. Minimally Invasive Local Therapies (Focal Therapy, HIFU, Cryotherapy)

Emerging focal therapies aim to treat PCa while sparing as much normal prostatic tissue and surrounding structures as possible. These include high-intensity focused ultrasound (HIFU), cryoablation, irreversible electroporation (IRE), and other organ-preserving approaches. Because they do not remove the entire prostate or seminal vesicles, and typically have less impact on neurovascular structures compared to radical surgery, the ejaculatory function can often be preserved [72,73]. Approximately 70% or more of patients treated with true focal therapy retain antegrade ejaculation, and aspermia is uncommon in reported series [74]. Likewise, because the testes are untouched and not exposed to systemic therapy, sperm production is generally unaffected, aside from transient effects of the acute treatment stress [75,76]. Some studies note a temporary mild oligospermia or asthenospermia after HIFU or cryotherapy, possibly due to inflammatory responses or short-term rises in temperature, but semen parameters often normalize by ~12 months post-treatment [77].

The endocrine axis remains intact (no induced hypogonadism) [78]. Therefore, in principle, men treated with focal therapies have a high likelihood of retaining natural fertility if cancer control is achieved. However, it is important to recognize that long-term fertility outcomes in this group are not well studied—many patients undergoing focal therapy are older or may have already completed family planning, and sample sizes in reports are small [79]. Clinicians should still engage in fertility discussions even for focal therapy candidates, to clarify that while fertility preservation is likely, it is not guaranteed and that sperm banking remains an option if there is any uncertainty or if the patient prefers to hedge against potential fertility loss. Monitoring of post-treatment semen quality can be offered to confirm preservation of fertility potential.

### 3.7. Emerging Systemic Therapies and Their Potential Effects on Male Fertility

Several novel systemic treatments for PCa have become available, including radiopharmaceuticals, immunotherapy, and targeted agents. These therapies have less established track records regarding fertility, and evidence on their reproductive side effects is sparse. It should be noted that any fertility impacts of the following agents are based mainly on theoretical mechanisms or limited observations and thus remain largely speculative pending further research [62]. Patients receiving these therapies should be counseled in general terms about potential risks to fertility, but definitive conclusions cannot yet be drawn from current data [80].

#### 3.7.1. Radium-223 Dichloride (Xofigo)

Radium-223 is an alpha-emitting radiopharmaceutical used to treat metastatic castration-resistant PCa (mCRPC) with bony metastases [81]. It targets areas of increased bone turnover, delivering localized radiation to bone metastases. Because Radium-223’s action is bone-specific, testicular exposure is minimal. Nevertheless, the product labeling cautions that radiation could theoretically impair spermatogenesis, and patients are typically advised to use contraception during and after treatment as a precaution [82]. Most men who receive Radium-223 are already on ADT (and thus already azoospermic and hypogonadal), making it difficult to isolate any additional effect of Radium-223 on fertility [83]. To date, no clinical studies have specifically evaluated sperm counts or fertility outcomes in men treated with Radium-223. There have been no reports of new-onset infertility beyond what ADT causes, and no documented cases of genetic defects in offspring. Thus, any gonadotoxic effect remains theoretical, and no direct evidence of fertility harm exists in humans so far.

#### 3.7.2. Sipuleucel-T (Provenge)

Sipuleucel-T is a dendritic cell-based immunotherapy used in mCRPC [84]. There are no published human data on its effects on spermatogenesis, sex hormones, or fertility. As treatment is typically administered to men already receiving ADT [83], isolating any additional fertility impact is challenging. No evidence to date suggests that sipuleucel-T adds any further reproductive toxicity beyond the underlying ADT. It does not appear to affect sexual or ejaculatory function directly, aside from any effects related to improved or worsened cancer status.

#### 3.7.3. PARP Inhibitors (Olaparib, Rucaparib)

Poly(ADP-ribose) polymerase (PARP) inhibitors are used in PCa patients who have certain DNA repair gene mutations (e.g., BRCA2) and have mCRPC [85]. These oral agents target tumor cell DNA repair pathways. There are currently no human studies examining sperm quality, hormone levels, or fertility outcomes with PARP inhibitor therapy in men. However, based on their mechanism, there are theoretical concerns: PARP inhibitors could induce DNA damage in rapidly dividing cells (including spermatogonia), and the drug (or its metabolites) might be present in semen. For this reason, clinical guidelines recommend that men on PARP inhibitors use effective contraception during treatment and for at least 3–6 months afterward [86].

#### 3.7.4. Lutetium-177–PSMA-617 (Pluvicto)

^177^Lu–PSMA-617 is a targeted radioligand therapy that delivers beta-particle radiation to cells expressing prostate-specific membrane antigen (PSMA), used in advanced mCRPC [87]. This treatment involves an injectable radiolabeled molecule that binds to PCa cells and irradiates them. Like Radium-223, patients receiving ^177^Lu-PSMA are almost invariably on ADT, so baseline fertility is already compromised. The additional radiation exposure to the testes from circulating ^177^Lu is generally low but not zero [88]. Animal studies on similar radioligand therapies indicate potential dose-dependent testicular effects, such as reduced spermatogenesis, when radionuclides are not fully cleared. In humans, no formal studies have yet evaluated the impact of ^177^Lu-PSMA therapy on fertility or sperm parameters. Given its mechanism, there is a theoretical risk of gonadal radiation that could impair spermatogenesis or cause DNA damage in sperm [89]. To be cautious, men are advised to use contraception during and for a period after therapy (often 6 months) and, if feasible, to cryopreserve sperm prior to treatment. Thus far, no fertility-related adverse events have been reported in clinical trials of ^177^Lu-PSMA, but the sample typically includes older men with closed family plans.

## 4. Mechanisms of Fertility Impairment in Prostate Cancer Treatments

PCa treatments adversely affect male fertility through multiple, often interrelated mechanisms. Each therapeutic modality targets distinct components of the male reproductive system, resulting in variable patterns and degrees of reproductive impairment. Beyond the treatment-specific effects outlined in Section 3, these mechanisms may be categorized more broadly into direct gonadal injury, endocrine disruption, ejaculatory dysfunction, oxidative damage to germ cells, and psychosocial sequelae. A comprehensive understanding of these overlapping pathways is essential for assessing the full scope of treatment-induced infertility and for guiding appropriate fertility preservation strategies. To synthesize these concepts visually, Figure 1 provides a schematic overview of the four principal mechanistic domains—testicular damage, ejaculatory dysfunction, oxidative stress, and psychosocial sequelae—through which prostate cancer therapies may impair fertility. Each pathway represents a distinct, yet often interrelated, contributor to male reproductive compromise, collectively highlighting the need for multidisciplinary assessment and intervention.

Prostate cancer therapies impair male fertility through four principal mechanisms: testicular damage, ejaculatory dysfunction, oxidative stress, and psychosexual sequelae. These mechanisms—either independently or in combination—converge to reduce spermatogenic capacity, hinder sperm delivery, damage sperm DNA, or reduce sexual activity, ultimately leading to infertility.

### 4.1. Testicular Damage and Hypogonadism

PCa therapies impair spermatogenesis through both direct testicular injury and secondary hormonal (endocrine) effects. Radiation-induced scatter to the testes is a key direct mechanism: the testes are among the most radiosensitive organs, with spermatogonial stem cells particularly vulnerable to ionizing radiation [90]. Even low incidental doses (~0.15 Gy) can reduce sperm counts, and exposures >0.3 Gy may cause temporary azoospermia [91]. Higher cumulative doses progressively deplete germ cells, with doses of 2–3 Gy ablating spermatocytes and 4–6 Gy eliminating spermatids [91]. Spermatogenic output typically declines 2–3 months after radiation, coinciding with the depletion of maturing germ cells, cohorts, and nadir sperm counts, which occur around 4–6 months post-treatment [92]. Recovery, if it occurs, is dose-dependent and protracted: after exposures <1 Gy, sperm counts might recover within ~1 year; after 2–3 Gy, recovery might take 2–3 years; and doses ≥4–6 Gy are frequently associated with irreversible azoospermia [91,93]. Modern EBRT techniques have reduced—but not eliminated—testicular exposure. In one prospective cohort, 84% of patients exceeded the threshold for oligospermia and 65% exceeded that for azoospermia; 10% received >2 Gy, associated with permanent sterility [90].

Mechanistically, radiation damages the DNA of developing germ cells, causing double-strand breaks and chromosomal aberrations that trigger p53-mediated apoptosis of spermatozoa. Surviving germ cells may suffer sub-lethal DNA damage, contributing to sperm aneuploidy or fragmentation (hence the increased DNA fragmentation seen post-radiation) [94]. Additionally, radiation can harm the supporting Sertoli and Leydig cells: doses above ~1 Gy to testes can impair Leydig cell function, leading to lowered testosterone and potential hypogonadism [95]. Leydig cells are relatively radio-resistant, but cumulative scatter doses >2 Gy have been linked to late-onset endocrine failure [96]. In long-term follow-up, men receiving EBRT alone demonstrated testosterone levels approximately 27% lower than age-matched controls, along with elevated FSH and LH, indicating compensated or overt primary hypogonadism [95].

Chemotherapeutic agents likewise cause direct testicular damage. Alkylating agents and platinum compounds (though not commonly used in PCa) are notoriously spermatotoxic. The taxane-based regimens used in PCa cause germ cell apoptosis, as evidenced by rises in FSH (reflecting reduced inhibin B from Sertoli cell-germ cell interactions) and testicular volume loss during therapy [97]. Histologically, chemotherapy can lead to maturation arrest or complete aplasia of the germinal epithelium. Some stem spermatogonia may survive and resume activity post-treatment, but often in an erratic or incomplete fashion [98].

Hormonal therapies (ADT) induce profound hypogonadism, which is essentially a functional form of testicular “shutdown.” By suppressing gonadotropins, ADT halts intratesticular testosterone production needed for spermatogenesis, leading to seminiferous tubule involution. Testicular biopsies after extended ADT show small, soft testes with atrophy of germinal epithelium and only Sertoli cells lining the tubules (a “Sertoli cell-only” syndrome) [99]. The extent of recovery after ADT varies. While testosterone and sperm output may return within 6–12 months in younger men receiving short-term ADT, especially when used neoadjuvantly, prolonged therapy (>18–24 months) is associated with incomplete or absent recovery of both endocrine and spermatogenic function [100]. Data from prospective cohorts indicate that only ~43% of men treated with 3 years of ADT recover eugonadal testosterone levels within five years of cessation, compared to ~87% recovery in those receiving short-term ADT or none at all [100]. The cumulative duration of therapy is a critical determinant of reversibility, with each additional month prolonging the time to hormonal normalization.

In summary, testicular injury and hormonal suppression are fundamental mechanisms by which radiation, chemotherapy, and ADT impair fertility.

### 4.2. Ejaculatory Dysfunction

Nearly all curative treatments for prostate cancer disrupt the normal ejaculatory process, thereby preventing sperm delivery even if some sperm production persists. Radical prostatectomy causes complete anejaculation through removal of the seminal vesicles and bilateral transection of the vas deferens [101]. Radiotherapy induces fibrosis and damage to the ejaculatory ducts and accessory glands, resulting in dry ejaculation or significantly reduced semen volume in the majority of men over time [102]. ADT and systemic agents contribute by reducing prostatic and seminal secretions through glandular atrophy and by diminishing orgasmic muscular contractions due to autonomic effects and hypogonadism [51]. Even high-dose chemotherapy, although not directly targeting the ejaculatory structures, can occasionally cause temporary ejaculatory failure via autonomic neuropathy or systemic fatigue [103].

Ejaculatory dysfunction encompasses anejaculation (complete absence of ejaculate) and aspermia (absence of sperm in any released fluid), both of which are distinct from erectile dysfunction yet have profound implications for fertility. In radical prostatectomy, ejaculatory dysfunction stems from the surgical removal of seminal fluid-producing structures and disruption of the vas deferens and ejaculatory ducts [104]. Radiotherapy can induce fibrosis or damage to the ejaculatory ducts and seminal vesicles, resulting in dry ejaculation or significantly reduced volume in the majority of men over time [105]. ADT and systemic therapies contribute by reducing the prostatic and seminal fluid secretions (through glandular atrophy) and diminishing the muscular contractions of orgasm due to autonomic nervous system effects and low androgen levels, culminating in little or no ejaculate [106]. Even treatments like high-dose chemotherapy that do not anatomically target reproductive structures can cause temporary ejaculatory failure in some cases, likely through autonomic neuropathy or extreme fatigue (though this is less common than the other modalities) [107].

In some cases, occult retrograde ejaculation may occur due to functional incompetence of the bladder neck following radiation or surgical disruption. This condition may go unrecognized unless post-orgasmic urinalysis is performed. It is crucial to differentiate ejaculatory dysfunction from erectile dysfunction; patients may report preserved orgasmic sensation but absent seminal emission [108].

### 4.3. Oxidative Stress and DNA Damage

Beyond gross anatomical and hormonal effects, PCa therapies can impair fertility at a molecular level by inducing oxidative stress and genetic damage in germ cells [109]. Radiotherapy is a prime culprit: irradiation of the testes (or even of the pelvic area with scatter reaching the gonads) generates reactive oxygen species (ROS) and causes direct DNA double-strand breaks in spermatogonial cells [110]. The resulting oxidative damage to sperm DNA can lead to increased rates of sperm DNA fragmentation in ejaculate samples post-treatment [111]. High levels of DNA fragmentation have been associated with reduced fertility and higher miscarriage rates [112]. Likewise, chemotherapeutic agents can alkylate DNA or interfere with DNA replication in spermatocytes and spermatogonia, leading to mutations or chromosomal aberrations in sperm. Even if some sperm survive, they may carry this subcellular damage [113].

Oxidative stress in the testes can also be induced by ADT indirectly. Low testosterone states may disrupt the balance of pro-oxidant and antioxidant systems in the testes; some studies in animals have shown increased oxidative markers in the testes under prolonged GnRH agonist therapy [114]. Moreover, inflammation secondary to tissue injury (e.g., radiation orchitis) can further increase ROS locally [115].

The long-term consequences of therapy-induced genetic damage are an area of active investigation. One concern is the health of offspring conceived after treatments [116]. Currently, evidence is somewhat reassuring: children of cancer survivors (including those treated for testicular cancer or Hodgkin lymphoma with chemo/radiation) generally do not show higher rates of genetic abnormalities than the general population. However, specific data in the PCa context (especially with newer therapies) are limited [117]. The risk of transmitting mutations acquired in sperm due to therapy is considered low, but patients often ask about this. Guidelines usually advise waiting a certain period after gonadotoxic therapy (e.g., 6–12 months) before attempting conception, partly to allow clearance of damaged sperm and turnover to a new cohort of spermatogenesis [118].

From a fertility standpoint, oxidative and DNA damage reduces the fertilizing potential of sperm and could impair embryo development. In vitro, sperm from men post-chemotherapy have shown higher rates of DNA fragmentation, which correlates with lower IVF success risk [119]. Antioxidant therapies are sometimes considered in men with high oxidative stress in semen (e.g., vitamin E and C supplementation), but their efficacy in the context of iatrogenic damage is unclear [120,121].

Overall, any man who has undergone radiotherapy or chemotherapy for PCa should have a semen analysis evaluating not just count but also DNA integrity if he plans to use his ejaculated sperm for reproduction, as high fragmentation might prompt consideration of using testicular sperm extraction (which sometimes yields less DNA-damaged sperm) or other approaches.

### 4.4. Psychosexual Sequelae Affecting Reproduction

Beyond direct physiological insults to reproductive organs and gametes, PCa treatments frequently give rise to psychological and sexual side effects that indirectly impair fertility [122]. As outlined in Section 3.1, Section 3.2 and Section 3.3, many men experience loss of libido, erectile dysfunction, and anejaculation as a result of treatment. These changes not only have physical implications but also profound psychosocial repercussions that can reduce or eliminate a couple’s capacity or desire to attempt conception. In effect, psychosexual sequelae serve as a modifiable yet often overlooked mechanism of infertility in PCa survivors [123].

The diagnosis and treatment of PCa can precipitate depression, anxiety, and altered self-image [124]. These feelings are amplified by treatment-induced sexual dysfunction. For example, men on ADT frequently report near-total loss of sexual interest and activity; one longitudinal study found up to 93% of men on long-term ADT ceased sexual intercourse entirely, citing lack of desire and erectile difficulties [125]. Even men who retain some erectile function after surgery or radiation often struggle with changes in orgasmic sensation (e.g., “dry orgasm” after RP or RT) and feel a sense of loss or emasculation associated with infertility. In surveys, the majority of PCa survivors report decreased sexual confidence (76%) and reduced sexual satisfaction after treatment [126]. Such internalized distress can lead to avoidance of sexual intimacy, thereby removing any chance of natural conception. Younger survivors who have not yet started or completed their families may feel hopeless or fear passing on a predisposition to cancer, causing some to consciously abandon their reproductive goals [127].

It is important to emphasize that these psychosexual factors are indirect yet significant contributors to infertility. Even when viable sperm might still be present (for instance, a man has some residual spermatogenesis after radiation), if he and his partner are not engaging in intercourse due to psychological or relationship strain, natural conception will not occur. Furthermore, if a couple is considering assisted reproduction, psychosexual health can influence their willingness to pursue such interventions. Emotional stress, marital strain, and reduced intimacy may prevent couples from following through with fertility treatments that could otherwise be an option [128,129].

Encouragingly, many PCa survivors do maintain interest in intimacy and can adapt with appropriate support. This underscores the importance of integrated care that addresses psychosexual wellness alongside oncologic outcomes [130]. Open communication with healthcare providers, referral to sexual counselors or psycho-oncology specialists, and use of interventions such as PDE5 inhibitors or other sexual aids can improve sexual function or satisfaction, thereby improving the chances of conception (or at least the couple’s quality of life) [131,132,133]. For example, couples counseling and sexual therapy can help partners navigate changes in sexual function and find new ways to maintain intimacy and pursue fertility, such as scheduling intercourse when libido is higher or exploring assisted reproductive techniques without intercourse.

From a fertility standpoint, addressing psychosexual sequelae means treating the whole patient: providing mental health support, sexual rehabilitation, and coping strategies as part of survivorship care. Studies indicate that men who receive psychosexual support report lower anxiety and better quality of life [134]. Tools such as validated questionnaires (e.g., the International Index of Erectile Function for sexual function, or the Fertility Distress Scale) can be used to screen for issues. Psycho-oncology programs often offer interventions tailored to cancer survivors’ sexual health—for instance, cognitive-behavioral therapy focusing on sexual confidence or peer support groups where men can share experiences and solutions [135]. Integrating these support tools is critical: a systematic review of psychosexual care in prostate cancer survivorship found significant unmet needs and highlighted that structured interventions and communication can greatly assist patients. As a practical measure, involving a psychologist or certified sex therapist early in the recovery phase can help mitigate the cascade where sexual dysfunction leads to relationship problems and ultimately a decision not to pursue family-building [136].

In conclusion, psychosexual sequelae constitute an important, modifiable mechanism of fertility impairment in PCa patients. Addressing these issues through counseling, medical management of sexual dysfunction, and emotional support can not only improve a patient’s quality of life but also potentially restore the opportunity or desire to have children.

## 5. Fertility Preservation Strategies and Clinical Counseling

As survival outcomes for PCa continue to improve, attention has increasingly shifted toward long-term quality of life, including the preservation of reproductive potential. Although PCa predominantly affects older men, a growing proportion of patients are being diagnosed at younger ages or before completing their families. Historically, fertility preservation in men with PCa has been underemphasized—often due to assumptions by providers that patients are too old or not interested in future childbearing [137]. However, contemporary clinical guidelines now advocate for early, proactive counseling about potential treatment-related infertility. This discussion should ideally occur at the time of diagnosis, prior to the initiation of any therapy with known gonadotoxic effects [138].

### 5.1. Sperm Cryopreservation Before Treatment

Sperm cryopreservation remains the cornerstone of fertility preservation for men undergoing PCa treatment and should be offered prior to any potentially gonadotoxic intervention, including radiotherapy, chemotherapy, radical prostatectomy, or prolonged ADT [139,140]. Collection is typically achieved via masturbation, with samples cryopreserved in liquid nitrogen for indefinite storage. In cases where ejaculation is not possible due to pain, psychological distress, or obstruction, the procedure may be adapted by performing collection under mild sedation or in specialized settings [141].

International guidelines, including those from the American Society of Clinical Oncology (ASCO) and the European Association of Urology (EAU), recommend that all men of reproductive age be counseled regarding the risk of infertility and offered sperm banking before initiation of therapy [142,143]. Although multiple ejaculates are ideal—typically 2–3 collected 48–72 h apart—to increase the quantity and quality of stored sperm, even a single sample can be clinically useful when time is limited [144,145].

Cryopreservation remains feasible and effective even in midlife patients, despite age-associated declines in semen quality. Post-thaw motility is typically reduced; however, the viability of cryopreserved sperm is sufficient for use in assisted reproduction. A recent meta-analysis involving over 23,000 cancer patients demonstrated that approximately 28% of individuals who used their banked sperm achieved pregnancy, and ~20% had a live birth, primarily through in vitro fertilization (IVF) and intracytoplasmic sperm injection (ICSI) [146]. Despite this, the overall utilization rate of cryopreserved sperm remains low (∼9%)—often due to evolving personal circumstances, alternative reproductive decisions, or spontaneous recovery of fertility [146]. Nevertheless, the psychological benefit and reassurance associated with sperm banking are well documented, even if samples are not ultimately used [147].

In men with PCa, baseline semen quality may be compromised. One institutional series reported a median sperm motility of only ~5% among men banking sperm before treatment (median age: 57 years), yet nearly all were able to successfully cryopreserve viable gametes [137]. Even when semen parameters are suboptimal, cryopreservation remains worthwhile, as viable sperm can be later utilized via ICSI, which requires only a minimal number of functional spermatozoa.

Importantly, the time window between diagnosis and initiation of definitive PCa therapy may be narrow. However, short delays of several days to accommodate sperm banking are generally considered oncologically safe and should be facilitated when possible [142,143]. Overall, sperm cryopreservation is a simple, low-risk, and widely accessible intervention that significantly enhances the probability of biological fatherhood and should be systematically integrated into the pre-treatment counseling of eligible PCa patients. A visual summary of the clinical decision-making pathway for fertility preservation in prostate cancer is presented in Figure 2.

### 5.2. Testicular Sperm Extraction (TESE) and MicroTESE

In PCa survivors who did not undergo sperm banking prior to treatment or who have developed irreversible anejaculation or azoospermia post-therapy, surgical sperm retrieval represents a viable fertility-preserving option. Testicular sperm extraction (TESE) involves the direct retrieval of sperm from the seminiferous tubules through a minimally invasive biopsy of the testicular parenchyma. A more advanced technique, microdissection TESE (microTESE), utilizes high-magnification microscopy to selectively identify areas of active spermatogenesis, thereby enhancing sperm yield while minimizing tissue removal [148,149].

Surgical retrieval is particularly indicated in men whose reproductive tract has been anatomically or functionally compromised by curative treatment modalities. Radical prostatectomy invariably results in loss of antegrade ejaculation due to disruption of the vas deferens and removal of seminal vesicles. Similarly, pelvic radiotherapy may damage the ejaculatory ducts and accessory glands, often resulting in functional aspermia. Long-term ADT, through suppression of the hypothalamic–pituitary–gonadal axis, leads to testicular atrophy and severe suppression of spermatogenesis [150]. In such contexts, ejaculate-based fertility is typically not recoverable, and TESE may be the only means of retrieving viable spermatozoa.

Although robust, large-scale studies in post-PCa patients are lacking, case reports and small series demonstrate that spermatogenesis may persist in isolated testicular foci even after intensive oncologic therapy. Notably, sperm retrieval has been successful in men treated with combinations of EBRT, ADT, and even bilateral orchiectomy, albeit with variable efficiency [151,152]. However, outcomes are influenced by several factors, including cumulative gonadotoxic exposure, age, and the duration of hormonal suppression.

Timing considerations are critical. In patients previously treated with ADT, a treatment-free interval—if oncologically permissible—may improve the probability of spermatogenic recovery. However, recovery is often incomplete after long-term suppression (>12 months) [153]. When proceeding with sperm retrieval, microTESE is generally preferred in the post-treatment setting, as spermatogenesis may be sparse and heterogeneously distributed [154]. Sperm retrieved intraoperatively may be used fresh for assisted reproduction or cryopreserved for future use.

Coordination with a fertility specialist is essential, particularly when TESE is performed in conjunction with a partner’s oocyte retrieval cycle. In men who have undergone bilateral orchiectomy, sperm retrieval is not feasible, and alternative family-building options must be considered. Nonetheless, in selected PCa survivors, TESE or microTESE remains a viable and clinically valuable intervention that may enable biological paternity in the absence of ejaculated sperm [155].

### 5.3. Use of Assisted Reproductive Technologies (ART)

ART represents the cornerstone of fertility restoration for PCa survivors, particularly those relying on cryopreserved or surgically retrieved sperm. The two primary modalities are IVF and ICSI. While IVF involves co-incubation of motile sperm with oocytes in vitro, ICSI entails the direct injection of a single spermatozoon into the cytoplasm of a mature oocyte [156].

In the context of PCa, ICSI is generally preferred due to its high efficacy in cases of compromised semen parameters. Post-treatment sperm—whether obtained via testicular extraction or thawed from cryopreservation—frequently exhibit reduced motility, count, or morphological integrity. ICSI enables successful fertilization even with limited or poor-quality sperm, making it the optimal technique for most PCa survivors [157].

A recent meta-analysis involving cancer patients who utilized banked sperm demonstrated that ICSI significantly outperforms both IVF and intrauterine insemination (IUI) in terms of success rates [146]. Clinical pregnancy rates per cycle were approximately 34% for ICSI, compared to 24% for IVF and 9% for IUI. Similarly, live birth rates were highest with ICSI, with a pooled delivery rate of ~23% per cycle [146]. These findings highlight the prevalence of male-factor infertility in this population and reinforce ICSI as the most reliable ART approach.

Although ICSI is typically the first-line method, IUI may be considered in select cases—such as when sperm quality is excellent and sufficient motile sperm were banked. However, the lower success rates and the need for repeated cycles limit its applicability [146]. In rare instances where spermatogenesis and ejaculatory function recover post-treatment (e.g., after a short course of ADT), spontaneous conception or IUI may be feasible. Nonetheless, in patients treated with modalities such as radical prostatectomy, natural conception remains virtually impossible due to the anatomical absence of ejaculate [158].

Despite these limitations, ART offers encouraging prospects for biological fatherhood in PCa survivors. Numerous reports confirm that sperm stored before treatment can remain viable for many years, and even sperm retrieved from post-treatment testes can result in successful fertilization and healthy offspring when combined with ICSI [159].

Optimal outcomes require a multidisciplinary approach, involving close collaboration between urologists, reproductive endocrinologists, and embryologists. Counseling should also address the potential need for donor sperm or alternative reproductive strategies, such as adoption, in cases where autologous sperm retrieval fails [155]. Nevertheless, for many men, ART enables the fulfillment of reproductive goals despite the profound effects of cancer therapy on natural fertility.

### 5.4. Counseling and Ethical Considerations

Fertility preservation counseling is an essential component of comprehensive PCa care. All men, regardless of age or disease stage, should be informed about the risk of treatment-induced infertility and the available preservation strategies before initiating therapy [142]. Current guidelines from ASCO and the EAU emphasize that fertility discussions must occur early—preferably at diagnosis—and be documented as part of informed consent [142]. In addition to sperm banking, patients should be informed about adjunctive medical and lifestyle-based interventions that may support fertility recovery or optimize residual spermatogenic function when present, including antioxidant supplementation, hormonal therapies, and empirically guided medical treatments [160].

The approach should be individualized, guided by the patient’s own reproductive goals and values. While some patients may decline fertility preservation due to age, prognosis, or personal preference, others—especially those with younger partners or future parenting desires—may opt for sperm banking despite advanced disease [161,162]. Surveys indicate that many men regret not being offered fertility preservation, and a substantial proportion would have chosen to bank sperm had they been counseled adequately [163].

Effective counseling must include clear, treatment-specific information: for instance, the irreversible loss of ejaculation after radical prostatectomy, the potential for DNA damage and aspermia following radiation, and spermatogenic suppression from ADT. Patients interested in preserving fertility should be referred promptly to reproductive specialists, as sperm cryopreservation can typically be arranged without delaying oncologic treatment [142].

From an ethical standpoint, failure to inform patients about fertility risks may result in decisional regret and psychological distress. Evidence shows that men who are offered fertility preservation experience reduced anxiety and greater satisfaction, even if they ultimately do not use their stored sperm [164,165]. Proactive discussions are associated with improved long-term quality of life and a greater sense of control during cancer treatment [166,167].

Informed consent for fertility preservation must also address legal considerations, particularly the disposition of stored gametes in the event of death or incapacity. Patients should explicitly document whether posthumous use is permitted, as local laws often prohibit such use without prior written consent [168]. Clinics must ensure that patients understand and complete these directives at the time of banking.

Additional ethical issues include the fate of unused samples and the financial burden of storage. Men should be counseled on storage duration, options for disposal or donation, and potential long-term costs. Equitable access is a concern, as financial barriers may prevent some patients from utilizing fertility preservation. Where possible, clinicians should assist with insurance navigation or refer patients to financial assistance programs or national oncofertility initiatives [169].

Finally, provider-related barriers—such as lack of time, knowledge, or comfort discussing fertility—must be addressed. Oncology teams should receive training on fertility preservation and implement streamlined referral pathways to reproductive specialists. Multidisciplinary survivorship programs incorporating reproductive health are strongly recommended to ensure consistent and equitable counseling [170].

In summary, comprehensive counseling is both ethically and clinically imperative. It ensures that patients are informed, that their choices are respected, and that those who wish to preserve fertility are provided with timely and appropriate options. Figure 3 illustrates a structured, time-based approach to fertility preservation across the prostate cancer treatment continuum, outlining key decision points, recommended actions, and supportive measures. This visual framework can help clinicians incorporate fertility discussions more systematically into routine oncologic practice, thereby enhancing the consistency and quality of care.

## 6. Emerging Research Needs and Gaps

Despite growing awareness of oncofertility issues, significant gaps remain in our understanding and implementation of fertility preservation for men with PCa. Identifying and addressing these gaps can guide future research and improve clinical practice.

### 6.1. Absence of Standardized Fertility Preservation Protocols in Prostate Cancer

Despite increasing awareness of survivorship issues, fertility preservation in PCa remains inconsistently addressed. Unlike pediatric and adolescent oncology, standardized oncofertility pathways for men with PCa are lacking [171]. As a result, fertility counseling is often omitted or delayed, especially in patients outside the traditionally reproductive age range.

Currently, there is no consensus on which PCa patients should be systematically referred for fertility preservation, nor are there standardized algorithms for integrating fertility counseling into treatment planning. Existing clinical guidelines (e.g., ASCO) advocate for pre-treatment discussions on infertility risks, yet their implementation in PCa care is variable. The development of unified, evidence-based protocols—including criteria for sperm cryopreservation, timing of referral, and management of cryopreserved specimens—is needed. Multidisciplinary coordination among urologists, oncologists, and reproductive specialists is essential to ensure that fertility preservation becomes a routine component of PCa care, particularly in men with curative intent and potential parenthood goals [172].

### 6.2. Limited Long-Term Data on Offspring Health

Data on the health outcomes of offspring conceived after PCa treatment remain sparse [173]. Most available evidence derives from studies of male cancer survivors across various tumor types. These studies suggest only a marginally increased risk of congenital anomalies, with absolute risks comparable to the general population [174,175]. However, they often lack treatment-specific detail and are not PCa-focused, limiting their applicability.

Moreover, long-term outcomes such as offspring development, fertility, cancer predisposition, or epigenetic alterations remain understudied [176]. Although sperm DNA damage is a recognized consequence of radiotherapy and systemic therapy, its potential transgenerational impact is unclear. There is a pressing need for PCa-specific cohort studies and registries to track offspring health outcomes following treatment-conceived pregnancies, particularly those involving cryopreserved or surgically retrieved sperm [113]. Until more robust data emerge, clinicians should communicate that available evidence is reassuring but incomplete.

### 6.3. Lack of Predictive Biomarkers for Post-Treatment Fertility

Currently, fertility prognostication in PCa relies on surrogate indicators such as semen analysis and reproductive hormone levels (FSH, LH, and testosterone) [177,178]. While useful in assessing testicular function, these parameters do not reliably predict post-treatment fertility potential. While these indicators can signal gonadal damage (for instance, a rising FSH suggests spermatogenic failure and Sertoli cell injury [148]), they are not specific predictors of a patient’s future ability to father children.

Emerging investigations have evaluated alternative markers, including inhibin B and anti-Müllerian hormone (AMH), as potential indicators of spermatogonial reserve [179,180]. Additionally, sperm DNA fragmentation assays and molecular profiling of testicular tissue are under exploration. However, no biomarker has been validated to forecast the likelihood of fertility recovery following PCa treatment.

This lack of predictive tools impedes personalized counseling. All patients must currently be advised of fertility risks in general terms, without a means to stratify individual prognosis. Prospective studies correlating pre- and post-treatment biological markers with reproductive outcomes (e.g., natural conception, ART success) are required. The identification of reliable fertility biomarkers would enable risk-adapted preservation strategies and facilitate individualized decision-making.

### 6.4. Precision Medicine and Artificial Intelligence in Fertility Risk Stratification

The application of precision medicine and artificial intelligence (AI) in oncofertility is a nascent but promising domain. Current fertility risk assessments rely on broad estimates based on treatment modality and age, without accounting for inter-individual variability. Integrating genomic, proteomic, and baseline reproductive data into fertility prediction models may allow clinicians to more accurately assess gonadotoxic risk. Machine learning algorithms trained on large, multicenter datasets could generate individualized risk profiles, analogous to existing nomograms used in oncology [181,182].

Such tools may assist in early identification of high-risk patients and facilitate timely intervention. Moreover, precision-based approaches could support the refinement of fertility-sparing techniques—such as testicular shielding during radiotherapy or treatment modifications in men with high reproductive potential. Future directions include building PCa-specific databases incorporating long-term reproductive outcomes and validating AI models for clinical use. While still exploratory, these technologies offer an opportunity to shift from generalized to personalized fertility counseling, enhancing the integration of reproductive health into survivorship care [183,184].

## 7. Conclusions

PCa treatments frequently compromise male fertility through a range of mechanisms, including anatomical disruption, hormonal suppression, DNA damage, and ejaculatory dysfunction. These effects are especially consequential for younger patients or those with ongoing reproductive aspirations. Despite the increasing incidence of early-onset PCa and rising survivorship rates, fertility preservation remains inconsistently integrated into standard care. Evidence reviewed herein underscores the significant risk of treatment-induced infertility across surgical, radiotherapeutic, hormonal, and chemotherapeutic modalities and highlights the necessity of early, proactive counseling and intervention.

To address these challenges, a multidisciplinary model involving urologists, oncologists, reproductive endocrinologists, and mental health professionals is essential. Fertility preservation should be recognized as a core quality-of-life domain in PCa survivorship, rather than an ancillary concern. Incorporating standardized protocols, refining patient selection, expanding access to preservation technologies, and investing in research on predictive biomarkers and offspring outcomes are critical steps forward. Systematic attention to fertility in PCa care not only supports patient autonomy and psychosocial well-being but also aligns with modern principles of personalized, holistic oncology.

## Figures and Tables

**Figure 1 jpm-15-00360-f001:**
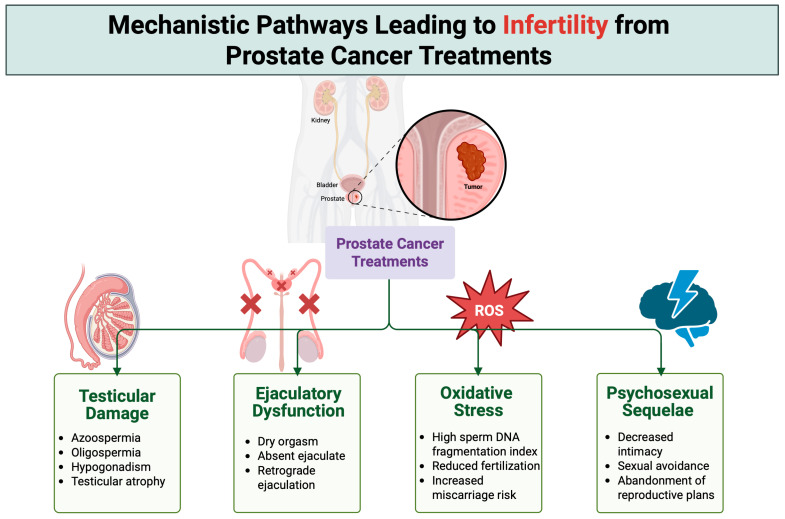
Mechanistic pathways of infertility induced by prostate cancer treatments.

**Figure 2 jpm-15-00360-f002:**
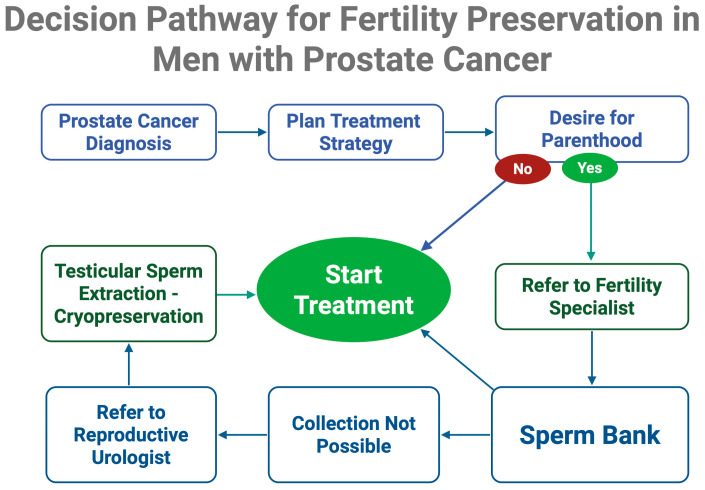
Fertility preservation pathway for men newly diagnosed with prostate cancer. Early assessment of reproductive goals and referral to a fertility specialist or reproductive urologist allows timely cryopreservation prior to initiating gonadotoxic therapy. Created in BioRender. Kaltsas, A. (2025) https://BioRender.com/yf9kkv6.

**Figure 3 jpm-15-00360-f003:**
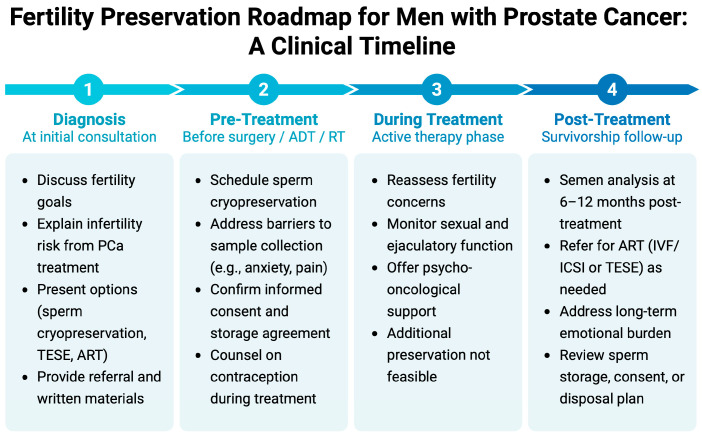
Fertility preservation roadmap for men with prostate cancer: a clinical timeline. Visual representation of counseling and intervention steps across four key phases—diagnosis, pre-treatment, active treatment, and survivorship—highlighting practical recommendations for sperm preservation, psychosocial support, and reproductive planning. Created in BioRender. Kaltsas, A. (2025) https://BioRender.com/3gq95jh.

**Table 1 jpm-15-00360-t001:** Summary of the effects of prostate cancer treatments on male fertility across key domains: ejaculatory function (EF), sperm parameters (SP), hormonal effects (HE), and fertility recovery potential (FRP). The table includes established and emerging therapies, with relevant citations reflecting recent literature. Recommendations for fertility preservation (e.g., sperm banking, TESE) are based on the likelihood of irreversible reproductive impairment.

Treatment	Ejaculatory Function (EF)	Sperm Parameters (SP)	Hormonal Effects (HE)	Fertility-Recovery Potential (FRP)
Radical Prostatectomy [34,35,36,37,38]	100% anejaculation (seminal vesicles + vas deferens excised)	• Aspermia • Testicular spermatogenesis intact	• Serum T largely unchanged • Mild ↑ FSH/LH possible	Natural conception impossible → micro-TESE ± ICSI required
Radiotherapy (EBRT ± brachytherapy) [39,40,41,42,43,44]	80–90% develop dry ejaculation or markedly ↓ volume on long-term follow-up	• ↓ Sperm count and motility • ↓ Semen volume • Risk of azoospermia • ↑ Sperm DNA fragmentation	Scatter ≥ 1 Gy may cause lasting Leydig cell damage and mild hypogonadism	Partial recovery if ≤ 0.5 Gy; ≥1–2 Gy ⇒ high chance of permanent infertility → cryopreserve before Tx
Androgen-Deprivation Therapy (ADT) [45,46,47,48,49,50,51,52]	Functional aspermia; profound ↓ libido and ejaculation frequency	• Virtually universal azo/oligospermia seminiferous tubular atrophy	Castrate T (<50 ng/dL); suppressed (GnRH agonist) or rebound (anti-androgen mono) LH/FSH	Young men on ≤6–12 mo may recover within a year; prolonged/older ⇒ often irreversible; bank sperm first
Chemotherapy (taxanes, cabazitaxel, etc.) [53,54,55,56,57,58,59,60,61,62]	Anatomical EF preserved but diminished by fatigue/neuropathy; occasional autonomic anejaculation	• ↑ azoospermia rates • ↑ DNA damage • ↓ inhibin B • ↑ FSH	• Usually transient ↓ T • Permanent hypogonadism is uncommon	Highly unpredictable—some regain counts after years; many remain infertile; pre-Tx cryopreservation essential
Next-generation AR pathway inhibitors (enzalutamide, apalutamide, darolutamide, abiraterone) [63,64,65,66,67,68,69,70,71]	With concomitant ADT: EF already absent; as monotherapy: mild ↓ ejaculate volume, libido	Human data is sparse; animal and mechanistic data predict ↓ spermatogenesis despite normal/elevated serum T	ADT-combined: castrate T; mono-Rx: serum T ↑ (pituitary disinhibition) but intratesticular T ↓	Reversible in principle after discontinuation, but no systematic human follow-up; advise sperm banking
Focal/organ-preserving local therapies (HIFU, focal cryotherapy, IRE, laser) [72,73,74,75,76,77,78,79]	~70% retain antegrade EF; aspermia is uncommon	Testes untouched; transient mild oligo–asthenospermia can normalize by 12 mo	Endocrine axis unaffected	High likelihood of natural conception; long-term fertility studies still limited; monitor semen post-Tx
Emerging systemic agents Radium-223 Sipuleucel-T PARPi ^177^Lu-PSMA [80,81,82,83,84,85,86,87]	Additional EF loss is minimal (patients are usually on background ADT)	No dedicated human data; potential for DNA damage (PARPi, ^177^Lu-PSMA)	Generally dictated by concurrent ADT; Radium-223; and ^177^Lu-PSMA may expose testes to low-dose radiation	Theoretical or unknown; regulatory guidance: counsel and cryopreserve when feasible

Abbreviations: EF: ejaculatory function; SP: sperm parameters; HE: hormonal effects; FRP: fertility recovery potential; ADT: androgen deprivation therapy; AR: androgen receptor; ARPI: androgen receptor pathway inhibitor; GnRH: gonadotropin-releasing hormone; ICSI: intracytoplasmic sperm injection; TESE: testicular sperm extraction; HIFU: high-intensity focused ultrasound; IRE: irreversible electroporation; PARPi: poly (ADP-ribose) polymerase inhibitor; ^177^Lu PSMA: lutetium-177-labeled PSMA ligand therapy; T: testosterone; LH: luteinizing hormone; FSH: follicle-stimulating hormone; mo: months; Tx: treatment; Rx: therapy; ↑: increased; ↓: decreased; ±: with or without; ⇒: leads to/associated with.

## Data Availability

Not applicable.

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
