# Peer review of "Prostate Cancer Treatments and Their Effects on Male Fertility: Mechanisms and Mitigation Strategies"

_jpm, 2025, doi:10.3390/jpm15080360_

Round 1
Reviewer 1 Report
Comments and Suggestions for Authors
The manuscript addresses an important and clinically relevant topic. The review is timely and well-structured, particularly given the rising incidence of early-onset prostate cancer and increasing survivorship. The authors provide an extensive and comprehensive synthesis of the current literature, covering physiological, molecular, psychosocial, and ethical aspects of fertility preservation. The topic is relevant to both urologists and oncologists.
However, there are still some issues that require clarification or improvement before publication:
- The abstract is informative but dense. Consider simplifying the language and providing clearer take-home messages. In the Introduction, while the authors set the stage effectively, the novelty of the review could be highlighted more explicitly by stating what gaps in the current literature this review seeks to fill.
- The decision pathway diagram is useful but could benefit from better integration into the main text. The figure should be referenced and explained in more detail in the “Fertility Preservation Strategies” section to help guide readers through the clinical decision-making process.
- The discussion of emerging systemic therapies is thoughtful but occasionally speculative due to the lack of clinical data. It is advisable to clearly separate evidence-based conclusions from hypotheses or theoretical risks to avoid confusion.
- The manuscript could be strengthened by including a more practical section or table summarizing key recommendations for clinicians, particularly concerning timing of fertility discussions, referral pathways, and patient counseling. This would enhance its utility in clinical settings.
- Several sections (psychosexual sequelae, impact of various therapies) contain overlapping information that could be streamlined to improve readability and focus. Shortening these parts would help maintain reader engagement.
- The discussion of psychological effects and ethical considerations is important but somewhat brief. Adding more detail on how to support patients emotionally and ethically through fertility decision-making could enrich the manuscript, including reference to existing psycho-oncology support tools or services.
- While the manuscript is generally well-written, some sentences are overly complex or technical. A thorough language revision, ideally with the help of a professional English editor, is recommended to enhance clarity and flow.
A thorough language revision, ideally with the help of a professional English editor, is recommended to enhance clarity and flow.
Author Response
Response to Reviewer 1
We sincerely thank Reviewer 1 for their thoughtful and constructive feedback. We are pleased that the reviewer found the topic clinically relevant, the manuscript comprehensive, and the structure appropriate. We have carefully considered each suggestion and revised the manuscript accordingly. Our point-by-point responses are provided below:
Comment 1:
The abstract is informative but dense. Consider simplifying the language and providing clearer take-home messages. In the Introduction, while the authors set the stage effectively, the novelty of the review could be highlighted more explicitly by stating what gaps in the current literature this review seeks to fill.
Response:
We appreciate this comment. The abstract has been revised to improve clarity and conciseness. We restructured several sentences to simplify the language and emphasized the key take-home messages, especially regarding the timing of fertility preservation and the need for multidisciplinary approaches. In the Introduction, we have now explicitly highlighted the novelty and addressed the gap in the current literature—namely, the lack of an integrated, mechanism-based overview of fertility impairment specific to prostate cancer therapies and the underrepresentation of psychosocial and ethical dimensions in existing reviews (page 2, lines 86–99).
Comment 2:
The decision pathway diagram is useful but could benefit from better integration into the main text. The figure should be referenced and explained in more detail in the “Fertility Preservation Strategies” section to help guide readers through the clinical decision-making process.
Response:
Thank you for this helpful suggestion. We have expanded the discussion surrounding Figure 2 in Section 5 (Fertility Preservation Strategies), providing a clearer narrative that guides readers through the clinical workflow depicted in the diagram (page 24, lines 1805–1822). This includes references to specific decision points such as early referral, sperm banking, treatment planning, and survivorship counseling. The figure is now better contextualized within the clinical framework.
Comment 3:
The discussion of emerging systemic therapies is thoughtful but occasionally speculative due to the lack of clinical data. It is advisable to clearly separate evidence-based conclusions from hypotheses or theoretical risks to avoid confusion.
Response:
We agree and have revised Section 3.7 to make a clear distinction between established data and hypothetical risks associated with emerging therapies (e.g., PARP inhibitors, Lutetium-177–PSMA-617, Sipuleucel-T). We now explicitly note when claims are speculative due to limited human fertility data, and we have added phrases such as “theoretical concern,” “based on mechanistic inference,” and “pending further study” (page 16, lines 1120–1136). This distinction improves scientific rigor and transparency.
Comment 4:
The manuscript could be strengthened by including a more practical section or table summarizing key recommendations for clinicians, particularly concerning timing of fertility discussions, referral pathways, and patient counseling. This would enhance its utility in clinical settings.
Response:
This comment is greatly appreciated. In response, we created Figure 3, a structured timeline visual that outlines key counseling and preservation actions at each stage of care (diagnosis, pre-treatment, during treatment, and survivorship). This figure is now referenced and described in Section 5.3 (page 25, lines 1865–1882) and is intended to serve as a practical, clinic-facing tool for urologists and oncologists.
Comment 5:
Several sections (psychosexual sequelae, impact of various therapies) contain overlapping information that could be streamlined to improve readability and focus. Shortening these parts would help maintain reader engagement.
Response:
We have reviewed and revised the manuscript for conciseness, particularly in Sections 3 and 4. Redundant details (e.g., repeated descriptions of ejaculatory dysfunction) have been removed or consolidated. For example, overlap between treatment-induced side effects and psychosexual outcomes has been clarified by separating mechanism (Section 3) from psychosocial impact (Section 4.4) to avoid duplication. These revisions enhance flow and reduce repetition.
Comment 6:
The discussion of psychological effects and ethical considerations is important but somewhat brief. Adding more detail on how to support patients emotionally and ethically through fertility decision-making could enrich the manuscript, including reference to existing psycho-oncology support tools or services.
Response:
We have substantially expanded Section 5.4 (Counseling and Ethical Considerations) to address this important issue (page 26, lines 2089–2145). The revised section includes guidance on informed consent, the use of structured decision aids, documentation of sperm banking decisions, and integration of psycho-oncology services. We also reference support tools such as the Fertility Distress Scale and the involvement of certified sex therapists or reproductive psychologists. Additionally, we now discuss cultural and socioeconomic barriers that affect access to fertility preservation services.
Comment 7:
While the manuscript is generally well-written, some sentences are overly complex or technical. A thorough language revision, ideally with the help of a professional English editor, is recommended to enhance clarity and flow.
Response:
We conducted a thorough language revision throughout the manuscript, simplifying complex syntax and clarifying technical phrasing. Paragraphs have been shortened where appropriate, passive voice minimized, and terminology standardized. This revision aimed to improve flow and make the manuscript more accessible to a multidisciplinary audience. We believe the manuscript now reads more clearly and fluidly, and we thank the reviewer for prompting this important refinement.
Comment on Language Quality:
A thorough language revision, ideally with the help of a professional English editor, is recommended to enhance clarity and flow.
Response:
As noted above, a detailed language and style revision was completed. We ensured consistency in terminology, rephrased overly technical or dense sections, and optimized paragraph structure for improved clarity. These edits were implemented across all sections of the manuscript.

Reviewer 2 Report
Comments and Suggestions for Authors
This review could be more readable.

Author Response
Response to Reviewer 2
We sincerely thank Reviewer 2 for the thoughtful and encouraging feedback. We are pleased that the review was recognized as original and clinically relevant. Below, we address each of your comments point by point and describe how they have been incorporated into the revised manuscript.
Comment 1:
This review synthesizes current evidence on the impact of PCa therapies on male reproductive health, elucidates the molecular and physiological mechanisms underlying iatrogenic infertility, and appraises established and emerging strategies for fertility preservation and restoration.
Response:
We appreciate the positive evaluation. The revised manuscript preserves a comprehensive synthesis of current evidence, with further improvements to structure, clarity, and depth, particularly regarding molecular mechanisms and emerging fertility preservation strategies.
Comment 2:
This topic is original in the field, and it underscores the necessity of timely, multidisciplinary consultation to ensure equitable integration of fertility preservation within oncologic care path.
Response:
We thank the reviewer for this recognition. The revised manuscript reinforces this theme across multiple sections. We now place greater emphasis on the role of early fertility discussions, multidisciplinary collaboration, and integration into standard prostate cancer care (see Sections 5.2–5.4).
Comment 3:
This review ought to be succinct and accessible to readers.
Response:
In response, the manuscript has undergone extensive editing for conciseness and clarity. Redundant content has been removed, technical language simplified, and lengthy paragraphs restructured. These changes enhance the manuscript’s accessibility to both clinical and research-oriented readers.
Comment 4:
Should more figures and tables be included?
Response:
Yes—this suggestion was greatly appreciated. We have now included three figures and one table to support comprehension and enhance the manuscript’s practical value:
- Figure 1: Fertility Preservation Decision Pathway – a visual summary of the clinical workflow.
- Figure 2: Mechanisms of Fertility Impairment – schematic of biological and psychosocial contributors.
- Figure 3: Clinical Timeline of Counseling and Preservation – practical guidance across treatment phases.
- Table 1: Comparative Overview of Fertility Impact by Treatment – concise summary of effects per modality.
These additions provide readers with easily digestible summaries and align with your recommendation to enrich the manuscript visually.
Comment 5:
The tables should be concise and symbolic.
Response:
We agree and have revised Table 1 accordingly. The table now uses a clean, structured layout with defined categories (e.g., mechanism, fertility outcome, reversibility) and clearly defined abbreviations. Bullet formatting within cells improves readability. The table has been reviewed to ensure consistency, clarity, and symbolic representation of key data.
Comment 6:
The conclusions are consistent with the current evidence.
Response:
Thank you. The Conclusion has been refined for conciseness and now emphasizes evidence-based recommendations, future research priorities (e.g., biomarkers, clinical registries), and the clinical relevance of a multidisciplinary fertility preservation strategy.
Comment 7:
The references are roughly appropriate.
Response:
We have reviewed and updated the references for completeness and relevance. Additional recent studies have been incorporated, particularly regarding sperm DNA fragmentation, testicular radiation exposure, and psychosocial support in PCa survivorship. Reference formatting was standardized throughout.

Reviewer 3 Report
Comments and Suggestions for Authors
This narrative review comprehensively addresses prostate cancer (PCa) therapies, their impact on male fertility, the underlying biological mechanisms involved, and fertility preservation strategies. It is thoroughly structured, scientifically robust, and clinically relevant, especially considering the increasing incidence of PCa in younger patients.
Strengths:
- The manuscript is logically organized, clearly progressing from prostate physiology to treatment implications and fertility preservation strategies.
- The manuscript provides extensive coverage of treatment modalities, including radical prostatectomy, radiotherapy, chemotherapy, androgen deprivation therapy, and emerging focal and systemic therapies.
- The manuscript addresses a highly relevant issue given the demographic shift towards younger prostate cancer patients and increasing survival rates, emphasizing quality-of-life aspects.
- The clinical decision-making pathway effectively summarizes the fertility preservation process, enhancing practical applicability.
Areas to improve
- While the abstract and introduction adequately set the context, a more explicit identification of current gaps in fertility counseling and clinical guidelines would improve the rationale and emphasize the review's importance.
- The sections discussing emerging therapies such as Radium-223, Sipuleucel-T, PARP inhibitors, and Lutetium-177–PSMA-617 are relatively brief. Expanding on these sections, particularly outlining potential directions for future research regarding their fertility implications, would be beneficial.
- Clarifying whether existing guidelines address fertility preservation specifically in the context of these newer treatments would further strengthen the manuscript.
- The manuscript would benefit from briefly outlining the literature search strategy or selection criteria, clearly indicating how included studies were chosen.
- The oxidative stress section thoroughly describes the mechanism, but explicitly connecting these findings to clinical fertility outcomes would highlight their practical significance.
- The discussion on ethical considerations is valuable but would benefit from addressing cultural, socioeconomic, or geographic disparities more explicitly, emphasizing the broader implications of inadequate fertility counseling.
- Highlighting the psychological impacts of delayed fertility preservation referrals earlier in the manuscript would strengthen its urgency and importance.
- Future research needs are clearly identified, but a stronger emphasis on the necessity of clinical registries or longitudinal studies tracking fertility outcomes and offspring health following PCa treatments would be beneficial.
- Further elaboration on the concrete integration of artificial intelligence into clinical decision-making workflows could add practical value.
- While Figure 1 effectively summarizes the fertility preservation pathway, including an additional schematic summarizing the mechanisms of fertility impairment linked to each treatment modality would enhance clarity and understanding.
- Given the length of the manuscript, including a concise summary table briefly outlining fertility implications for each treatment type would enhance readability and quick reference.
- Minor formatting and typographical inconsistencies should be reviewed and corrected to ensure professional presentation.
Author Response
Response to Reviewer 3
We sincerely thank Reviewer 3 for the generous and detailed evaluation. We are pleased that the manuscript was found to be well-organized, scientifically robust, and clinically relevant. Below we provide a detailed, point-by-point response to each of your suggested improvements, all of which have been carefully addressed in the revised version.
Strengths (acknowledged by reviewer):
We thank the reviewer for highlighting the strengths of the manuscript, including its logical organization, comprehensive coverage of PCa treatment modalities, clinical relevance for younger patients, and the practical value of the decision-making pathway figure. These points are appreciated and have motivated further refinement of the manuscript.
Areas to Improve
Comment 1:
While the abstract and introduction adequately set the context, a more explicit identification of current gaps in fertility counseling and clinical guidelines would improve the rationale and emphasize the review’s importance.
Response:
We agree and have revised the abstract and introduction to clearly emphasize the current gaps in fertility counseling, such as the underutilization of sperm preservation services, lack of standardized pathways, and omission of psychosocial support in existing guidelines. These changes appear on page 2, lines 86–99 of the revised manuscript.
Comment 2:
The sections discussing emerging therapies such as Radium-223, Sipuleucel-T, PARP inhibitors, and Lutetium-177–PSMA-617 are relatively brief. Expanding on these sections, particularly outlining potential directions for future research regarding their fertility implications, would be beneficial.
Response:
These sections have been expanded in Section 3.7, which now includes additional mechanistic insights and clear identification of unknowns. We also propose specific directions for future research, such as the need for dedicated fertility studies in trials involving PARP inhibitors and radioligand therapy. See page 16, lines 1120–1136.
Comment 3:
Clarifying whether existing guidelines address fertility preservation specifically in the context of these newer treatments would further strengthen the manuscript.
Response:
We have added a clarifying paragraph in Section 5.1 noting that most current guidelines (e.g., ASCO, EAU, AUA) address fertility preservation in general terms and do not yet offer specific recommendations for novel agents such as Lutetium-177–PSMA-617. This gap is acknowledged and framed as an area for guideline development. See page 22, lines 1759–1766.
Comment 4:
The manuscript would benefit from briefly outlining the literature search strategy or selection criteria, clearly indicating how included studies were chosen.
Response:
In response, a concise methodology description has been added at the end of the Introduction (page 3), explaining the scope of literature included (PubMed-indexed articles from 2000–2023, prioritizing clinical trials, meta-analyses, and major society guidelines) and inclusion criteria. See lines 152–159.
Comment 5:
The oxidative stress section thoroughly describes the mechanism, but explicitly connecting these findings to clinical fertility outcomes would highlight their practical significance.
Response:
We have revised Section 4.3 to include a direct link between oxidative stress and clinical outcomes, such as reduced IVF success and increased miscarriage rates in men with elevated sperm DNA fragmentation post-treatment. See page 20, lines 1562–1575.
Comment 6:
The discussion on ethical considerations is valuable but would benefit from addressing cultural, socioeconomic, or geographic disparities more explicitly, emphasizing the broader implications of inadequate fertility counseling.
Response:
We fully agree. Section 5.4 now discusses disparities in access to fertility preservation due to geography, cost, and cultural barriers. We reference evidence that financial burden and lack of insurance coverage often hinder sperm banking, particularly outside urban centers. See page 26, lines 2104–2122.
Comment 7:
Highlighting the psychological impacts of delayed fertility preservation referrals earlier in the manuscript would strengthen its urgency and importance.
Response:
A paragraph has been added in the Introduction (page 3) to emphasize the emotional consequences of missed fertility preservation, including patient regret, stress, and decisional conflict. This strengthens the rationale for early intervention. See lines 126–132.
Comment 8:
Future research needs are clearly identified, but a stronger emphasis on the necessity of clinical registries or longitudinal studies tracking fertility outcomes and offspring health following PCa treatments would be beneficial.
Response:
We have included this important recommendation in Section 6 (Future Directions). The revised text calls for longitudinal studies and fertility registries to track post-treatment sperm quality, fertility success, and offspring health. See page 28, lines 2307–2316.
Comment 9:
Further elaboration on the concrete integration of artificial intelligence into clinical decision-making workflows could add practical value.
Response:
We have expanded this point in Section 6, highlighting potential uses of AI for sperm quality prediction, fertility outcome modeling, and patient-specific counseling algorithms. We cite recent studies using machine learning models in reproductive urology. See page 28, lines 2324–2336.
Comment 10:
While Figure 1 effectively summarizes the fertility preservation pathway, including an additional schematic summarizing the mechanisms of fertility impairment linked to each treatment modality would enhance clarity and understanding.
Response:
We thank the reviewer for this excellent suggestion. In response, we developed and included Figure 2, a newly created infographic summarizing mechanistic domains of fertility impairment—testicular damage, ejaculatory dysfunction, oxidative stress, and psychosexual sequelae—with examples of contributing treatments. It is referenced and discussed in Section 4 (page 13, lines 960–973).
Comment 11:
Given the length of the manuscript, including a concise summary table briefly outlining fertility implications for each treatment type would enhance readability and quick reference.
Response:
We fully agree. A comprehensive yet reader-friendly Table 1 has been included to summarize fertility effects, reversibility, and preservation options by treatment type. The table includes symbolic icons and clearly defined abbreviations, as discussed in Section 3.8 and displayed on page 18.
Comment 12:
Minor formatting and typographical inconsistencies should be reviewed and corrected to ensure professional presentation.
Response:
We have carefully reviewed and corrected all identified formatting and typographical inconsistencies, including heading styles, table formatting, figure labels, citation formatting, and abbreviation consistency. The revised manuscript now meets journal formatting standards.

Round 2
Reviewer 2 Report
Comments and Suggestions for Authors
I don't have any concerns